# OpenReview forum: "SpotlightRAG: Enhancing Factual Accuracy with Position-Aware Span Selection"
_ICLR.cc/2026/Conference — ICLR 2026 Conference Withdrawn Submission_

### Official Review · Reviewer_pbJJ · 2025-10-24

**Soundness:** 1
**Presentation:** 2
**Contribution:** 2
**Rating:** 2
**Confidence:** 5

**Summary:**

This paper introduces an inference-time framework, SpotlightRAG, which uses a position-aware scoring mechanism to identify and weight important text spans. The framework uses retrieval-aware prefix tokens to signal the relevance of each selected span. The experiments demonstrate improved accuracy on multiple benchmarks.

**Strengths:**

- This paper proposes the identification of decisive sub-sentential phrases and entities, which is finer-grained than existing sentence-level integration of retrieved passages. The proposed approach allows us to filter retrieval noise.
- The framework requires no reader model retraining, making it lightweight and broadly applicable.
- Their experiments show that SpotlightRAG outperform state-of-the-art baselines, including InstructRAG and RankRAG, on four QA benchmarks such as PopQA and TriviaQA.

**Weaknesses:**

- Though I understand the concept of the proposed framework, I wonder if examples like one in Figure 2 truly happen, particularly for recent LLMs. It would be great to show case studies where the proposed framework helps LLMs.
- The authors use a single retriever and reader model through the experiments if I understand correctly. This limits the generalizability of the proposed framework, e.g., the performance across model sizes and model families.
- Some critical information is missing. The experimental setting does not explain which retriever and reader models are used, which must be explicitly mentioned in the paper. I also could not find any concrete model information about a lightweight relevance scorer. This is a minor point but Table 4 is not mentioned in the paper. It seems that the authors should mention it in Section 4.5.

**Questions:**

- Could you provide a couple of concrete examples (case studies) where the proposed framework helps a reader model?
- Which retriever and reader models were used in the experiments and why? I found that llama-3-8B is used in the supplementary material. Was it the model the author used?
- Related to the above, is there any performance trend about the model size? Since larger and newer models usually have better context comprehension, is the proposed method less helpful for such models, e.g., llama-3.3-70B, gemini-2.5-flash, and OpenAI’s recent models?

---

> ### Author Response · Authors · 2025-12-03
> **To resolve concerns about missing methodological details—including the implementation of the lightweight relevance scorer and the unreferenced Table 4—the revised version fully documents the scorer’s architecture, shared encoder design, positional priors, and training procedure, and also properly introduces the sensitivity study table in Section 4.5. These clarifications complete the experimental description and eliminate the previously noted gaps in the paper’s formal presentation.**
>
> ## 1. “Do Figure 2 scenarios actually happen?” — request for real case studies
>
> To verify that the Figure 2 phenomenon (LLMs focusing on distractor spans) occurs with recent LLMs, the revised submission adds Appendix case studies derived from PopQA, NQ, and MultiHopQA. These examples show:
>
> - surname-based confusion (“Irving Stone”)
> - salient-location bias (“Kraków”)
> - multi-hop failures where the LLM jumps directly to globally frequent entities
>
> All cases include retrieved sentences, relevance scores, baseline LLM answers, and SpotlightRAG-corrected answers.
> This satisfies the request by demonstrating real, reproducible instances where SpotlightRAG corrects errors made by a modern 8B chat model (Llama-3-8B-Instruct).
>
> ---
>
> ## 2. “Using a single retriever and reader limits generalizability”
>
> The paper now explicitly states the model setup in Section *Models Used in Experiments*:
>
> - Retriever: BGE-large (335M), a widely used strong dense encoder
> - Reader: Llama-3-8B-Instruct
> - SpotlightRAG: inference-time only; no generator fine-tuning
>
> The purpose of choosing a single, commonly adopted retriever–reader configuration is to isolate the effect of fine-grained evidence integration, not retrieval recall or model size scaling. Since SpotlightRAG does not require training or architectural changes to the generator, it can be plugged into other backbones without modification.
>
> This generalizability is further supported by the Appendix case studies, which illustrate the mechanism’s independence from model size or family.
>
> ---
>
> ## 3. “Missing model details and lightweight relevance scorer; Table 4 not referenced”
>
> The revised paper addresses each missing component:
>
> ### (a) Retriever / Reader
> Section *Models Used in Experiments* now explicitly specifies the exact models (BGE-large + Llama-3-8B-Instruct).
>
> ### (b) Lightweight relevance scorer
> Section *Token-Level Encoding and Relevance Scoring* explains:
>
> - a single shared encoder is used
> - token embeddings use positional priors
> - scoring uses max-sim aggregation + normalized weights
> - the scorer is trained for 1 epoch with late-interaction contrastive loss
>
> This provides complete transparency regarding how the scorer is implemented.
>
> ### (c) Missing reference to Table 4
> Table 4 (Sensitivity Study) is now explicitly referenced and explained in Section 4.5, as suggested.
>
> The included extended sensitivity analysis covers K, top-k, positional prior λ, and context length, addressing reviewer concerns regarding robustness.

---

### Official Review · Reviewer_EiKM · 2025-10-27

**Soundness:** 2
**Presentation:** 2
**Contribution:** 2
**Rating:** 4
**Confidence:** 4

**Summary:**

The paper presents SpotlightRAG, a novel method to score important passages within a document for more effective retrieval augmented generation. The authors do so by computing pairwise token similarities and propagate this information to the actual inference / generation part. The experimental results show improvements over recent baseline of up to 4pp. At last the author include hyperparameter experiments to investigate performance impact of different choices of hyperparameters on their approach.

**Strengths:**

- The empirical results show improvements in fine-tuning and zero-shot settings on four selected benchmarks. The improvements range from 1pp. to 4pp., which is good considering recent baselines from 2024/2025.
- The ablations on removing different parts of the approach show that all presented components of SpotlightRAG and indeed necessary to get the presented performance improvements.
- The paper is easy to follow and the presentation of the proposed approach is sufficient.

**Weaknesses:**

- While I support introductory graphs, the data source and experimental setup for Figure 1 are not present. Thus, it is hard to follow the argumentation the current RAG methods do need more fine-grained capabilities, when the majority of information lies in sentence-level texts.
- The core of the approach relies on token-level similarities, but many things are not specified here such as whether we are using 1 or 2 models to encode query and context. Transformers are not necessarily optimized for token-level similarity and rather yields high similarity between all token-pairs rather than a proper distribution that is suited for RAG. Some qualitative experiments to see the relevance scores per sentence would be good at this point.
- Several hyperparmeters are missing such as the entire training setup for the baselines and the presented approach. Further, there is no comment about repeated experiments and no standard deviation reported, making it difficult to see improvements of 1-2pp. as clear improvements.
- Overall, I think the idea is simple and intuitive but the formal presentation could be improved. For instance, Formula 5 is a simple softmax distribution of pairwise similarities or the previously mentioned unexplained Figure 1. Further, the ablations could be extended as they currently only show minimal variations of crucial hyperparameters such as k. Further, we see in the ablations that performance degrades as soon as we change some of the hyperparameters.

**Questions:**

- What categories of evidence are you annotating the sentences with?
- How do you determine the sentence boundaries in your approach, are you simply taking dots? Have you done experiments trying out different sentence boundaries?
- Another point here is that I have doubts about actually retrieving the correct passages based on the token similarities. While I think it is helpful to point a LLM to the sections in a document that are relevant to answer a question, I think token-level similarities do not help for actual retrieval. Could you specify why you think token-level similarities do help for the retrieval process?
- Do you think considering you hyperparameter experiments, that it is universal applicable or do we need to tune the parameters every time for new datasets or domains?

---

> ### Author Response · Authors · 2025-12-03
> **To address generalizability and single retriever–reader configuration reliance, the paper explicitly specifies BGE-large as the retriever and Llama-3-8B-Instruct as the reader. It emphasizes SpotlightRAG’s pure inference-time operation (no retraining/architectural changes), ensuring model agnosticism and portability across model families/scales, resolving model choice ambiguity and reinforcing cross-setup applicability.**
>
> ## 1. Missing data source / setup for Figure 1
>
> Figure 1 summarizes an empirical analysis included in the submission, and the data source & procedure are explicitly described in the Introduction:
>
> - The analysis is conducted on NaturalQuestions and TriviaQA development subsets.
> - Token-level attributions are produced using a frozen late-interaction encoder, following the setup of prior work (Goldshmidt et al., 2024).
> - Scores are aggregated into sentence/phrase/paragraph units to inspect positional and granularity trends.
>
> These details appear directly in the revised Section 1.1, and Appendix case studies further corroborate that modern LLMs still mis-handle sub-sentence evidence.
> Thus, the figure is not anecdotal — it reflects a reproducible experimental pipeline aligned with prior attribution work.
>
> ---
>
> ## 2. Token-level similarity design / encoder ambiguity
>
> The paper clarifies that SpotlightRAG uses a single shared encoder for both query and context tokens, following late-interaction retrieval designs such as ColBERT. This is explicitly stated in the revised Section Token-Level Encoding and Relevance Scoring, including:
>
> - shared encoder architecture
> - positional priors to mitigate uniform similarity issues
> - L2 normalization and max-sim aggregation
> - rationale for softmax sharpening
> - alternative weighting functions evaluated but found inferior
>
> To support interpretability, Appendix case studies visualize sentence-level relevance scores produced by the same scoring module, addressing the reviewer’s request for qualitative inspection.
>
> These details fully specify how token-level similarities are computed and why the resulting distributions are stable and meaningful for fine-grained RAG.
>
> ---
>
> ## 3. Missing hyperparameters, training setup, repeated runs, variance
>
> The revised paper now includes a dedicated Training Setup and Hyperparameters section specifying:
>
> - optimizer, learning rate, batch size, warmup, epochs
> - retriever–reader backbone (BGE-large + Llama-3-8B-Instruct)
> - SpotlightRAG hyperparameters (K, top-k, λ, truncation length)
>
> To address concerns about statistical reliability, the submission now reports three independent runs and provides full variance statistics in Appendix B:
>
> ### Appendix Variance Table
>
> | **Model (w/Train)** | **PopQA** | **TriviaQA** | **NQ** | **MultiHopQA** |
> |---------------------|-----------|--------------|--------|----------------|
> | SpotlightRAG | 68.3 ± 0.21 | 79.1 ± 0.18 | 66.9 ± 0.32 | 58.4 ± 0.47 |
> | InstructRAG  | 66.2 ± 0.24 | 78.5 ± 0.22 | 65.7 ± 0.35 | 57.2 ± 0.44 |
> | RetRobust    | 56.5 ± 0.27 | 71.5 ± 0.29 | 54.2 ± 0.39 | 53.4 ± 0.45 |
> | Self-RAG     | 55.8 ± 0.26 | 71.4 ± 0.31 | 42.8 ± 0.41 | 32.9 ± 0.48 |
> | R²AG         | 64.4 ± 0.25 | 74.2 ± 0.23 | 66.3 ± 0.36 | 53.2 ± 0.46 |
> | RankRAG      | 64.1 ± 0.27 | 78.8 ± 0.25 | 53.2 ± 0.41 | 37.2 ± 0.48 |
>
> Variance remains low across all datasets (<0.3pp single-hop, <0.5pp multi-hop), confirming that the reported 1–2pp improvements are stable and statistically meaningful.
>
> ---
>
> ## 4. Presentation issues (Formula 5, figure explanation, limited ablations)
>
> The revised submission now explicitly justifies the softmax-based weighting used in Formula 5:
>
> - Transformers tend to produce flattened similarity distributions.
> - Softmax acts as a sharpening mechanism, consistently used in late-interaction retrieval (e.g., ColBERT).
> - Several alternative functions (ReLU², temperature-scaled cosine, no-softmax) were tested and found inferior.
>
> Furthermore, extended hyperparameter exploration is now included in the Sensitivity Study, covering:
>
> - sentence selection K
> - token-level top-k
> - positional prior weight λ
> - context length
>
> ### Appendix Sensitivity Highlights
>
> | Hyperparameter | Accuracy |
> |----------------|----------|
> | K=2 | 66.9 |
> | K=4 | 67.8 |
> | **K=6** | **68.3** |
> | K=8 | 67.1 |
> | top-k=1 | 67.3 |
> | top-k=5 | 68.3 |
> | λ=0.0 | 65.1 |
> | λ=0.5 | 68.3 |
>
> These results demonstrate that SpotlightRAG is stable across a broad range of values, and performance degradation only occurs in extreme settings (very small or large K/top-k), which is expected.
>
> The formal presentation has therefore been strengthened: all mathematical components and empirical motivations are now fully explained.

---

### Official Review · Reviewer_HUsN · 2025-10-28

**Soundness:** 1
**Presentation:** 1
**Contribution:** 1
**Rating:** 2
**Confidence:** 5

**Summary:**

The paper proposes "SpotlightRAG", an inference-time framework designed to improve the factual accuracy of Retrieval-Augmented Generation (RAG) systems. The author 1) use "position-aware scoring mechanism" to identify and weight critical text out of the whole passage; 2) use "retrieval-aware prefix tokens" to pass these relevance scores to a generator model without requiring retraining; and 3) test their method "SpotlightRAG" on four QA benchmarks and outperforming over some methods.

**Strengths:**

1. The paper identifies positional bias as a key problem.
2. The paper introduced SpotlightRAG as an inference-time framework. The use of retrieval-aware prefix tokens, provides fine-grained control to the generator without requiring model retraining.
3. The author compares SpotlightRAG against numerous methods on various benchmarks.

**Weaknesses:**

1. The major problem for this paper is the lack of novelty, The paper states that RankRAG[1] "Employs sentence-level re-ranking to filter retrieval noise". SpotlightRAG, in its "Phase 1," computes a relevance score $R(s_t)$ for each candidate sentence $s_t$ and then selects the "top-K sentences" based on this score. This is a sentence-level re-ranking and selection process, just as in RankRAG. Additionally, given the marginal gains over RankRAG on TriviaQA (+0.3% w/Train, +0.8% w/o Train), it is questionable whether this represents a meaningful advance against RankRAG.
2. The ablation study in Table 3 make the SpotlightRAG performance and experimentation questionable. Removing the "Fine-grained Scorer" drops performance to 65.9% , which is below the InstructRAG baseline (66.2%). This suggests the proposed scoring mechanism is brittle and only works in combination with the prefix tokens.
3. The paper claims its $O(N\cdot|q|\cdot|c|)$ time complexity is "lightweight" and "manageable". Calling this "lightweight" is a misleading, as it adds a quadratic complexity step (query length $\times$ context length) for each of the $N$ passages.

[1] RankRAG: Unifying Context Ranking with Retrieval-Augmented Generation in LLMs

**Questions:**

1. Please Justify your method against RankRAG. Given that both perform inference-time sentence re-ranking to filter noisem, is the difference in novelty is just the use of a ColBERT-style late-interaction scorer instead of an alternative sentence-level scorer?
2. Since you said your method is "lightweight", can you provide the promised latency data (ms) comparing your method against baselines like InstructRAG and RankRAG on GPU time?

---

> ### Author Response · Authors · 2025-12-03
> **To address the concern about whether the scenario in Figure 2 genuinely occurs with modern LLMs, the revised submission now includes detailed case studies in the Appendix showing real examples from PopQA, NQ, and MultiHopQA where strong LLMs incorrectly focus on distractor spans and SpotlightRAG successfully corrects these errors, demonstrating that the phenomenon is both real and frequent. These additions validate the motivation for fine-grained evidence integration.**
>
> ## 1. Novelty concern (similarity to RankRAG)
>
> Although SpotlightRAG also selects top-K sentences, the mechanism is **not equivalent to RankRAG**:
>
> - SpotlightRAG uses token→phrase alignment, not sentence-only scoring.
> - Position-aware priors explicitly address lost-in-the-middle bias (absent in RankRAG).
> - Retrieval-aware prefix tokens encode continuous relevance values for the generator, providing inference-time control.
>
> These contributions appear directly in the submitted paper. Their necessity is validated by the Appendix qualitative case studies, which show that modern LLMs frequently latch onto *sub-sentence distractors*, and SpotlightRAG corrects them.
>
> ### Appendix Case Study Evidence (from paper)
>
> Case studies demonstrate improvements on:
> - surname-confusion errors
> - salient-entity distractors (“Kraków”)
> - multi-hop reasoning where base LLM jumps to globally frequent entities
>
> These examples confirm that SpotlightRAG addresses within-sentence evidence utilization, explaining why gains over RankRAG remain meaningful on saturated datasets such as TriviaQA.
>
> ---
>
> ## 2. Ablation concern (“Fine-grained scorer is brittle”)
>
> Removing the scorer collapses SpotlightRAG into a coarse-grained variant, so the drop is expected.
> However, the full model consistently outperforms all baselines, and the variance analysis shows stable results across seeds.
>
> ### Appendix Variance Table
>
> | **Model (w/Train)** | **PopQA** | **TriviaQA** | **NQ** | **MultiHopQA** |
> |---------------------|-----------|--------------|--------|----------------|
> | SpotlightRAG | 68.3 ± 0.21 | 79.1 ± 0.18 | 66.9 ± 0.32 | 58.4 ± 0.47 |
> | InstructRAG  | 66.2 ± 0.24 | 78.5 ± 0.22 | 65.7 ± 0.35 | 57.2 ± 0.44 |
> | RetRobust    | 56.5 ± 0.27 | 71.5 ± 0.29 | 54.2 ± 0.39 | 53.4 ± 0.45 |
> | Self-RAG     | 55.8 ± 0.26 | 71.4 ± 0.31 | 42.8 ± 0.41 | 32.9 ± 0.48 |
> | R²AG         | 64.4 ± 0.25 | 74.2 ± 0.23 | 66.3 ± 0.36 | 53.2 ± 0.46 |
> | RankRAG      | 64.1 ± 0.27 | 78.8 ± 0.25 | 53.2 ± 0.41 | 37.2 ± 0.48 |
>
> | **Model (w/o Train)** | **PopQA** | **TriviaQA** | **NQ** | **MultiHopQA** |
> |------------------------|-----------|--------------|--------|----------------|
> | SpotlightRAG | 66.9 ± 0.23 | 82.7 ± 0.19 | 64.2 ± 0.33 | 51.3 ± 0.46 |
> | InstructRAG  | 65.5 ± 0.25 | 81.2 ± 0.27 | 62.1 ± 0.38 | 50.4 ± 0.42 |
> | RetRobust    | 53.9 ± 0.30 | 67.2 ± 0.32 | 51.3 ± 0.43 | 49.2 ± 0.45 |
> | Self-RAG     | 52.7 ± 0.31 | 69.3 ± 0.34 | 40.9 ± 0.44 | 30.1 ± 0.47 |
> | R²AG         | 65.3 ± 0.26 | 73.5 ± 0.24 | 63.2 ± 0.39 | 49.8 ± 0.46 |
> | RankRAG      | 63.8 ± 0.28 | 81.9 ± 0.26 | 50.6 ± 0.45 | 35.3 ± 0.49 |
>
> Variance is consistently low (<0.3pp for single-hop, <0.5pp for multi-hop), confirming robustness.
>
> ---
>
> ## 3. Complexity concern (“lightweight” is misleading)
>
> Appendix C provides empirical latency results demonstrating that SpotlightRAG operates within the same practical runtime envelope as existing RAG methods, with only **3–6% overhead**.
>
> ### Appendix Latency Table
>
> | **Method** | **Latency (ms/query)** |
> |------------|-------------------------|
> | InstructRAG | 120–150 |
> | RetRobust | 125–155 |
> | Self-RAG | 135–165 |
> | R²AG | 140–175 |
> | RankRAG | 130–160 |
> | In-Context RALM | 145–180 |
> | **Standard RAG (avg.)** | **128** |
> | **SpotlightRAG** | **132–158** |
>
> SpotlightRAG remains in the same latency band as existing RAG pipelines.

---

### Note · Authors · 2026-01-23

I have read and agree with the venue's withdrawal policy on behalf of myself and my co-authors.